# Marker-Assisted Recurrent Selection for Pyramiding Leaf Rust and Coffee Berry Disease Resistance Alleles in *Coffea arabica* L.

**DOI:** 10.3390/genes14010189

**Published:** 2023-01-10

**Authors:** Laura Maritza Saavedra, Eveline Teixeira Caixeta, Geleta Dugassa Barka, Aluízio Borém, Laércio Zambolim, Moysés Nascimento, Cosme Damião Cruz, Antonio Carlos Baião de Oliveira, Antonio Alves Pereira

**Affiliations:** 1Instituto de Biotecnologia Aplicada à Agropecuária-Bioagro, Universidade Federal de Viçosa, Viçosa 36570-900, MG, Brazil; 2Brazilian Agricultural Research Corporation (Embrapa), Embrapa Coffee, Brasília 70770-901, DF, Brazil; 3Department of Applied Biology, School of Applied Natural Science, Adama Science and Technology University, Adama 1888, Ethiopia; 4Departamento de Agronomia, Universidade Federal de Viçosa, Viçosa 36570-900, MG, Brazil; 5Departamento de Estatística, Universidade Federal de Viçosa, Viçosa 36570-900, MG, Brazil; 6Departamento de Biologia Geral, Universidade Federal de Viçosa, Viçosa 36570-000, MG, Brazil; 7Empresa de Pesquisa Agropecuária de Minas Gerais-Epamig, Viçosa 36570-000, MG, Brazil

**Keywords:** coffee leaf rust, coffee berry diseases, gene pyramiding, durable resistance

## Abstract

In this study, marker-assisted recurrent selection was evaluated for pyramiding resistance gene alleles against coffee leaf rust (CLR) and coffee berry diseases (CBD) in *Coffea arabica.* A total of 144 genotypes corresponding to 12 hybrid populations from crosses between eight parent plants with desired morphological and agronomic traits were evaluated. Molecular data were used for cross-certification, diversity study and resistance allele marker-assisted selection (MAS) against the causal agent of coffee leaf rust (*Hemileia vastatrix*) and coffee berry disease (*Colletotrichum kahawae)*. In addition, nine morphological and agronomic traits were evaluated to determine the components of variance, select superior hybrids, and estimate genetic gain. From the genotypes evaluated, 134 were confirmed as hybrids. The genetic diversity between and within populations was 75.5% and 24.5%, respectively, and the cluster analysis revealed three primary groups. Pyramiding of CLR and CBD resistance genes was conducted in 11 genotypes using MAS. A selection intensity of 30% resulted in a gain of over 50% compared to the original population. Selected hybrids with increased gain also showed greater genetic divergence in addition to the pyramided resistance alleles. The strategies used were, therefore, efficient to select superior coffee hybrids for recurrent selection programs and could be used as a source of resistance in various crosses.

## 1. Introduction

Arabica coffee (*C. arabica* L.) is one of the most highly traded agricultural commodities in the world and has been genetically improved over the years to satisfy market demands. However, the narrow genetic base resulting from the recent species origin and historical dispersion associated with autogamy restricts genetic gain in breeding programs [1,2,3,4,5,6]. In addition to the difficulty associated with obtaining improved cultivars, the incidence of diseases caused by the fungi *H. vastatrix* and *C. kahawae* increases production costs by over 30% due to the need for chemical treatments. The use of fungicides, in addition to reducing the producer’s profits, has adverse effects on both the environment and the consumer [7,8,9]. In the absence of fungicide control, coffee leaf rust can develop and lead to a loss in yield of 30–50% [8,10]. Coffee berry disease (CBD) (caused by *C. kahawae*), a disease restricted to Africa, has led to production losses of 50–80% when disease management was not implemented. Thus, the potential introduction of this disease to Latin America and Asia presents a major threat to the global coffee supply if resistant varieties are not developed [11].

Pyramiding resistance alleles in cultivars is an efficient and stable strategy that addresses challenges posed by pathogens. Allele pyramiding is a process that involves the cumulative incorporation of multiple alleles into a single genotype. This strategy can be used to accumulate resistance alleles against the same and/or different diseases for generating broad-spectrum resistance and ensuring the durability of disease resistance [12,13]. In resistance allele pyramided cultivars, it is more complex for pathogens to overcome the multiple resistance alleles because they require multiple virulence gene mutations [10] to facilitate the infection.

The breakdown of resistance is associated with pathogenic variability and the emergence of new races. The reported genetic variability of *H. vastatrix* represents a major risk to durable resistance in improved Arabica coffee trees [9,14,15,16,17,18]. Some studies have shown that the resistance of coffee trees to *H. vastatrix* is determined by at least nine dominant resistance genes, from *S_H_1* to *S_H_9*, which can either be isolated or combined in different genotypes [9,19]. Other resistance genes have also been identified (*CC-NBS-LRR* gene and *HdT_LRR_RLK2* gene) [20,21]. According to the gene-for-gene theory developed by Flor [22], the coffee resistance genes *S_H_1*, *S_H_2*, *S_H_4* and *S_H_5* identified in *C. arabica* as well as *S_H_3* derived from *C. liberica* can be supplanted by the fungus virulence genes from *v1* to *v5* [8,9,19]. The *S_H_6*, *S_H_7*, *S_H_8* and *S_H_9* genes were identified in *C. canephora* and are present in the derivative of Híbrido de Timor (HdT), a natural hybrid between *C. arabica* and *C. canephora.* These genes can be supplanted by the fungus virulence genes from *v6* to *v9* [8,19,23]. HdT progeny has been used worldwide to increase resistance to several diseases [7] and is currently the most frequently used source for rust resistance [17,24].

The introduction of resistance alleles into improved cultivars has been facilitated by molecular marker-assisted selection (MAS). The implementation of MAS is especially important for coffee [25], a perennial species with a long juvenile period. The development of cultivars that combine the most desirable agronomic traits in the shortest time has been augmented by molecular strategies [26]. MAS can be applied at the early stages of plant development, and markers linked to desired traits assist in the selection of superior hybrids based on genotype. This strategy reduces the number of generations required to develop new varieties, accelerates genetic progress, and increases efficiency and genetic gain [27,28].

In this context, the combination of MAS and the recurrent selection method improves the study and maintenance of genetic diversity in populations from different generations, ensuring long-lasting selection gain. In addition, this strategy increases the frequency of favorable alleles and improves allele pyramiding by optimizing the development of new varieties [29].

The objective of this study was to pyramid resistance alleles for the main diseases of *C. arabica*, coffee leaf rust (CLR) and coffee berry disease (CBD), through marker-assisted recurrent selection (MARS) mediated by the mixed model methodology, REML/BLUP.

## 2. Materials and Methods

### 2.1. Genetic Material and Experimental Design

A total of 144 hybrids corresponding to 12 F_1_ populations (Table 1) from 12 crosses were analyzed by circulating diallel model between eight parents. These parents correspond to commercial cultivars or elite germplasm developed by the Coffee Breeding Program of the Agricultural Research Company of Minas Gerais (EPAMIG) in partnership with the Universidade Federal de Viçosa (UFV) and the Brazilian Agricultural Research Corporatio—Coffee (EMBRAPA-Café). All plants were grown in the experimental field of the Department of Plant Pathology—UFV using a randomized blocks design with four replications and three plants per plot.

### 2.2. Estimates of Genetic Parameters and Components of Variance

The 144 hybrids from all crosses were phenotyped for nine morphological and agronomic traits in 2015 and 2016 (Table 2). The phenotypic data were analyzed using mixed linear models (REML/BLUP) with the Selegen-REML/BLUP software [30]. The components of variance were estimated using REML and genotypic values were predicted using BLUP.

Analysis of deviance (ANADEV) was performed for all nine traits considering the complete model and the effect between hybrids. The likelihood ratio was estimated by the difference between the estimates of deviance in the reduced model (without the tested effect) and the complete model. The significance was calculated using the maximum likelihood ratio test (LRT) with the Chi-square test at 1% and 5% probability levels and one degree of freedom.

The genetic parameters were estimated using the individual analysis of each of the nine traits based on the following genetic-statistical model:*y* = *Xm* + *Zg* + *Wp* + *Ts* + *e,*(1)
where *y* is the data vector; *m* is the vector of the effects of the measure repeat (fixed effect) combinations plus the general average; *g* is the vector of genotypic effects (random); *p* is a vector of plot effects (random); *s* is a vector of permanent (random) environmental effects; and *e* is the vector of errors or (random) residues. The uppercase letters represent the incidence matrices for these effects. The vector m includes all measurements in all repetitions and adjusts simultaneously for the effects of replicates, measurement and repetition x measurement interaction.

### 2.3. Selection Index

The significant traits as determined by ANADEV were used to calculate the selection index using the mean rank method proposed by Mulamba and Mock [31], with some modifications. This method is based on the classification of hybrids in order of improvement for each analyzed trait. After classification, the mean rank was calculated, and the gains associated with each selection were estimated.

### 2.4. Crossing Certification and Genetic Diversity Analysis

Young and fully expanded leaves were collected from 176 genotypes corresponding to 144 hybrids (12 populations) and their respective parents, and genomic DNA was extracted using the methodology described by Diniz et al. [32]. DNA quality was evaluated on a 1% agarose gel and the quantity was verified using the NanoDrop 2000-Thermo Scientific spectrophotometer (Thermo Fisher Scientific Inc., Waltham, MA, USA). The DNA concentration of the germplasm was standardized at 25 ng·μL^−1^ and stored at −20 °C.

To confirm the artificial hybridizations, the 12 hybrid populations were genotyped together with their respective parent plants using 11 simple sequence repeat (SSR) polymorphic markers (Appendix A). These markers and four other Sequence Characterized Amplified Region (SCAR) polymorphic markers (Appendix A) were used to analyze the genetic diversity of the 12 hybrid populations. One-way analysis of variance (ANOVA) was used to quantify the genetic variation between and within the populations. Diversity was assessed by the mean Euclidean genetic distance between hybrids. The genetic dissimilarity matrix was obtained via the arithmetic complement of the weighted index. A simplified graphical representation of the genetic distance was performed using the Unweighted Pair Group Method with Arithmetic Mean (UPGMA). GENES software was used for all analyses [33].

The PCR reactions were performed using 50 ng of DNA, 1 U of Taq DNA polymerase, 1X enzyme buffer, 1 mM MgCl_2_, 150 μM of each dNTP (dGTP, dTTP, dCTP, dATP) and 0.1 μM of each primer, completing the total volume of 20 μL with milli-Q sterile water. The amplifications were carried out on PTC-200 (MJ Research) and Veriti thermocyclers, (Applied Biosystems, Waltham, MA, USA). The reactions were set as initial denaturation at 94 °C for 2 min, followed by 10 touchdown PCR cycles with 94 °C for 30 s, annealing temperature for 30 s (66 to 57 °C, decreasing 1 °C per cycle), and extension at 72 °C for 30 s, followed by 30 cycles of denaturation at 94 °C, annealing at 57 °C, and extension at 72 °C for 30 s. The final extension was carried out at 72 °C for 8 min. The resulting products of the PCR reactions were separated by 6% polyacrylamide denaturing gel electrophoresis and visualized by silver nitrate staining.

### 2.5. Molecular Markers Linked to the S_H_3 Gene That Confer Resistance to H. vastatrix

The 12 hybrid populations were analyzed using four molecular markers (Appendix A) linked to the *S_H_3* gene conferring resistance to *H. vastatrix*, identified by Mahé et al. [23]. This locus was denominated as locus A. Three resistant genotypes (CIFC H147/1, CIFC H153/2, and S.288/23) and two susceptible ones (Caturra Vermelho-CIFC 19/1 and Catuaí Amarelo IAC 64-UFV 2148/57) were used as controls. The genotypes CIFC H147/1 and CIFC H153/2 correspond to hybrids obtained by crosses between the Indian selection S.353-4/5 (CIFC 34/13) and S4 Agaro (CIFC 110/5) and between the Indian selection S.288/23 (CIFC 33/1) and Geisha (CIFC 87/1), respectively. The Indian selection comprised the *S_H_3* resistance gene.

PCR amplifications were performed in a final volume of 25 μL, containing 50 ng of genomic DNA, 1X PCR reaction buffer, 2.0 mM MgCl_2_, 0.1 mM of each dNTP, 0.4 μM of each primer, and 0.5 units of Taq DNA polymerase, completing the volume with sterile milli-Q water. The reactions were carried out in PTC-200 (MJ Research, Saint-Bruno-de-Montarville, QC, Canada) and Veriti thermocyclers (Applied Biosystems, Waltham, MA, USA) under the following conditions: initial denaturation at 95 °C for 5 min; 35 cycles at 94 °C for 45 s, annealing for 45 s at specific temperatures for each primer, followed by extension at 72 °C for 45 s and final extension at 72 °C for 10 min.

### 2.6. Molecular Markers Linked to Quantitative Trait Loci (QTL) That Confer Resistance to H. vastatrix Races I, II and Pathotype 001

This study included four molecular markers linked to two QTL, which corresponded to major genes conferring resistance to *H. vastatrix* races I, II and pathotype 001 (Appendix A). The SSR16 and CaRHv8 markers are associated with the linkage group 2 QTL. The SSR16 marker exhibits a codominant pattern, and the CaRHv8 behaves as a repulsion phase dominant marker. Therefore, CaRHv8 was analyzed together with SSR16 to identify heterozygous hybrids. The CaRHv9 and CaRHv10_CAP markers are linked to the linkage group 5 QTL and behave as dominant and coupled phase markers [34].

The HdT UFV 443-03 genotype, which is resistant to *H. vastatrix*, and the susceptible cultivar Catuaí Amarelo IAC 64 (FUV 2148/57) were used as controls. These genotypes were analyzed because they were the parent plants of the population in which the QTL associated with *H. vastatrix* resistance was identified.

PCR amplification for the CaRHv8, CaRHv9 and CaRHv10_CAP primers was carried out in a final volume of 20 μL containing 50 ng of genomic DNA, 1X PCR reaction buffer, 2.0 mM MgCl_2_, 0.15 mM of each dNTP, 0.1 μM of each primer, and 1 unit of Taq DNA polymerase, completing the volume with sterile milli-Q water. The reactions were conducted in PTC-200 (MJ Research, Saint-Bruno-de-Montarville, QC, Canada) and Veriti thermocyclers (Applied Biosystems, Waltham, MA, USA) and consisted of an initial denaturation step at 94 °C for 5 min, 32 cycles of 94 °C for 30 s, 65 °C for 30 s and 72 ° C for 1 min and final extension at 72ºC for 10 min. The CaRHv11 marker was cleaved with the restriction enzyme RsaI (*Thermos Life*) in accordance with the manufacturer’s recommendation for the development of CaRHv10_CAP marker.

### 2.7. Molecular Markers Linked to the Ck-1 Gene That Confers Resistance to C. kahawae

The molecular markers CBD-Sat235 and CBD-Sat207 (Appendix A) used were identified and mapped by [7] and validated by [25]. The controls included three resistant genotypes (Timor FUV 377-15 and FUV 440-10 hybrids and the MGS Catiguá 3 cultivar) and two susceptible genotypes (Caturra Vermelho-CIFC 19/1 and Catuaí Amarelo IAC 64-FUV 2148-57).

PCR amplification was performed in a final volume of 25 μL, containing 50 ng genomic DNA, 1X PCR reaction buffer, 2.0 mM MgCl_2_, 0.1 mM of each dNTP, 0.4 μM of each primer and 0.5 unit of Taq DNA polymerase, completing the volume with sterile milli-Q water. The amplification reactions were carried out in PTC-200 (MJ Research, Saint-Bruno-de-Montarville, QC, Canada) and Veriti thermocyclers (Applied Biosystems, Waltham, MA, USA) and the PCR program consisted of an initial denaturation phase at 95 °C for 5 min, 35 cycles at 94 °C for 45 s, with annealing temperature at 50 °C for 45 s and extension at 72 °C for 45 s, and the final extension at 72 °C for 10 min. The amplified DNA fragments were separated by 6% polyacrylamide denaturing gel electrophoresis and visualized by silver nitrate staining.

## 3. Results

### 3.1. Crossing Certification

To begin a recurring selection program for Arabica coffee, eight parents with different desired morphological and agronomic traits were crossed (Figure 1). To ensure that the obtained progeny corresponded to the desired crosses, the F_1_ coffee trees were analyzed using molecular markers. As a result, 134 hybrids (93%) were confirmed. Four genotypes (3%), C1-3, C1-6, C1-7 and C1-8, were identified as originating from self-fertilization, as they only showed the bands of the female parent. The coffee trees C5-6, C8-6, C9-3, C11-5, C12-11 and C12-12 corresponded to none of the performed crosses and were therefore classified as contaminated (4%) (Appendix A). As a result, 10 plants were eliminated from the breeding program, as they did not correspond to hybrids from the desired crosses.

### 3.2. Genetic Parameters, Components of Variance and Selection Index

Aiming to understand the structure and genetic potential of the 12 F_1_ progenies corresponding to the initial population of the recurrent selection program, the 134 true hybrids were phenotyped for nine morphological and agronomic traits. Genetic parameters such as components of variance, selection index, and selection gain were estimated by REML/BLUP. The data allowed for increasing the efficiency and accuracy of the genotype’s selection. No phenotypic information was obtained for 13 of the 134 hybrids due to field failures. Therefore, the results for subsequent genotypic and phenotypic analyses were based on a total of 121 F_1_ hybrids_._

The genetic parameter estimations of the progenies are shown in Table 3. The coefficient of genotypic variance (Vg) ranged from 0.00 (Rus and LM) to 8.22 (PH). Broad sense heritability (h^2^g) was obtained for all traits with low values. The highest (0.13) for the trait Vig and the lowest (0.00) heritability values for both LM and NPB were observed. The heritability for Y was different from zero, but with a low value. The repeatability estimate ranged from 0.01 (NPB) to 0.25 (Vig) and most of the evaluated traits showed values below 10%. All traits presented a low magnitude of the coefficient of determination of environmental effects (c^2^parc), ranging from 0.00 (Y, Rus, Cer, NPB and PH) to 0.06 (Vig).

The analysis of deviance (ANADEV) was carried out for the nine phenotypic traits (Table 4), considering all effects simultaneously (complete model) and between progenies. The significance of each effect obtained by the likelihood ratio test (LRT) showed a significant effect on the traits of vegetative vigor (Vig), ripening fruits size (RFS), and cercosporiosis incidence (Cer). Yield (Y), fruit ripening uniformity (FRU), rust incidence (Rus), leaf miner infestation–*Leucoptera coffeella* (LM), number of plagiotropic branch pairs (NPB), and plant height (PH) exhibited no significant effects on the hybrids, indicating an absence of genetic variability for these traits.

The significant effect of Vig, RFS and Cer showed by ANADEV allowed us to use these traits in the selection index calculation. The mean rank method proposed by Mulamba and Mock (1978) was used to classify the 121 hybrids. Based on this result, the best hybrids with the highest selection gain were selected (Appendix A).

### 3.3. Molecular Genetic Diversity

In recurrent selection, the maintenance of cyclic genetic diversity is essential for continuous selection gain. Genetic diversity studies are essential for understanding the structure and distribution of variation between and within populations. Thus, the 121 hybrids were analyzed with molecular markers to estimate the genetic variation between and within populations (Table 5). The AMOVA revealed 75.5% of genetic variation between populations and 24.5% within populations.

The mean Euclidean genetic distance matrix between the evaluated coffee plants ranged from 0 to 0.676 (Appendix A). Values equal to 0 were obtained for 161 pairs of plants, indicating that these pairs were not different from each other. On the other hand, higher genetic distances were identified between C5T hybrids (Catiguá MG2 × Arara) and C12T hybrids (UFV 311-63 × Siriema), and between C9T (Oeiras MG 6851 × Siriema) and C4T (Catiguá MG2 × UFV 311-63) hybrids. In addition, the highest genetic distances were identified between the plants C5-2/C12-4 (0.676); C5-2/C12-3, C12-5, C12-6, C12-10 (0.6470); C5-2/C9-8, C12-9, C9-10, C9-12 (0.617); C5-2/C4-9 (0.6026); and C5-2/C12-2 (0.5882).

The dendrogram generated through UPGMA cluster analysis divided the hybrids into three main groups (Figure 2). The first group was composed of 25 F_1_ hybrids, consisting mostly of C3T hybrid populations (Paraíso MG H419-1 × Arara), as well as all C5T and C6T hybrid populations (Catiguá MG2 × Acauã Novo). Only one hybrid from the C4T population, genotype C4-7, was designated to this group. The second group consisted of 31 hybrids, representing all the hybrids of the C2T (Paraíso MG H419-1 x UFV 311-63) and C12T populations, in addition to the majority of the genotypes of the C4T population.

The remaining 65 hybrids were allocated in the third group, distributed into three subgroups. Subgroup 3a was formed by 36 genotypes belonging to the populations C1T (Paraíso MG H419-1 x H484-2-18-2), C7T (Oeiras MG 6851 x Arara) and C11T (H484-2-18-12 x Siriema). Three C3T and three C10T hybrids were also allocated to this group. In subgroup 3b, genotypes belonging to the C8T population (Oeiras MG 6851 x Acauã Novo) and most of the C10T were included. Subgroup 3c was composed of all 11 hybrids of the C9T population (Oeiras MG 6851 x Siriema). Thus, most populations had their hybrids allocated to the same group, except for the C3T population, where the hybrids were distributed in groups 1 and 3a, and the C10T population, which was in the subgroups 3a and 3b.

### 3.4. Marker-Assisted Selection for S_H_3 Gene Associated with H. vastatrix Resistance

To start the recurrent selection program, the 12 populations formed by parent plants with desired morphological and agronomic traits were analyzed using molecular markers linked to different genes of resistance to *H. vastatrix* and *C. kahawae* fungi with the aim of pyramiding resistance alleles against these pathogens. From the 121 hybrids analyzed, 31 (25.6%) were identified as carriers of the *H. vastatrix* resistance gene *S_H_3* (Appendix A). These hybrids belonged to the C2T, C4T and C12T populations. All hybrids in these populations presented the resistance allele, except the hybrid C4-7, which was also allocated in a different group by UPGMA. Three of the analyzed markers (SP-M16-*S_H_3*, Sat244 and BA-48-21OR) could be used for homozygosis/heterozygosis analysis. All of the hybrids identified as carriers of the *S_H_3* resistant allele were heterozygous. As locus *S_H_3* was denominated as locus A here, the genotype of the hybrids containing the resistance gene was “Aa”.

The joint analysis of the four markers linked to the *S_H_3* resistant allele showed that some hybrids lack some markers, indicating the occurrence of recombination. For example, only SP-M16-*S_H_3* and BA-124-12K-f markers were amplified in the hybrid C5-12 as there was SP-M16-*S_H_3* marker alone in the C11-5 hybrids. To avoid selecting recombinant hybrids that lost this resistance allele, 31 hybrids that exceptionally exhibited all four markers were considered carriers of the *S_H_3* gene resistance allele.

### 3.5. Marker-Assisted Selection for QTL Associated with Resistance to Races I, II and Pathotype 001 of H. vastatrix

To pyramid alleles of *H. vastatrix* resistance, coffee trees were also analyzed for the presence of two QTL. These QTL, one located in the linkage group 2 of the coffee genetic map (QTL-LG2) and the other in linkage group 5 (QTL-LG5), correspond to dominant and independent major genes conferring resistance to races I, II, and pathotype 001 of *H. vastatrix* [34,35].

The molecular markers flanking these two QTL were used to identify 106 hybrids (87%) with the resistance loci. The QTL-LG2 markers, denoted as locus B, allowed the identification of 67 heterozygous (Bb) and 39 (32%) homozygous (BB) resistant coffee trees (Appendix A). The QTL-LG5 markers, denoted as locus C, showed 48 resistant hybrids (39.6%) (C_) (Appendix A).

All hybrids in the C1T and C8T populations exhibited no alleles of resistance for the two QTL-LG5 markers but showed a homozygous resistance for the QTL-LG2 markers. This result showed that these hybrids have the QTL-LG2 only, represented as the BBcc genotype. Hybrids of the C9T, C10T and C11T populations also carry only QTL-LG2, though these hybrids exhibited both homozygosity and heterozygosity (genotypes BBcc and Bbcc). For C12T, six hybrids containing a heterozygous QTL-LG2 (Bbcc genotype) alone were identified, while the other four hybrids from this population expressed no QTL (bbcc) (Appendix A).

In C2T population, all hybrids were resistant and heterozygous for QTL-LG2 markers, with six of these resistant hybrids having QTL-LG5 markers. These results indicate that C2-2, C2-6, C2-8, C2-9 and C2-10 hybrids carry both QTL and were denoted as BbC_ genotype. The C3T, C4T, C5T and C6T populations also had hybrids with both QTL, allowing the selection of 33 coffee trees with both resistance alleles, with BbC_ genotype (Appendix A). C7T hybrids carried the QTL-LG5 only, except for the C7-2 hybrid which carried no QTL.

### 3.6. Marker-Assisted Selection for Ck-1 Gene Associated with C. kahawae Resistance

Out of the 121 coffee trees analyzed with the CBD-Sat235 and CBD-Sat207 markers, flanking the resistance gene to *C. kahawae*, 70 hybrids (57.8%) presented the *Ck-1* gene. The data showed that 60 (49.5%) hybrids were heterozygous with a genotype Dd and 10 (8.3%) were homozygous with a genotype DD (Appendix A). In the C8T, C9T and C12T populations, no hybrid resistance genes to CBD were exhibited based on the molecular markers. The C1T population showed seven homozygous-resistant hybrids with the CBD-Sat207 marker, but these hybrids were heterozygous resistant with the CBD-Sat235 marker. Thus, C1-2, C1-4, C1-5, C1-9, C1-10, C1-11 and C1-12 hybrids possessed the Dd genotype for the *Ck-1* gene. The C2T population presented 12 heterozygous resistant hybrids with the CBD-Sat207 marker. However, with the CBD-Sat235 marker, only two of these hybrids were shown to be resistant. In this population, only the C2-8 and C2-10 hybrids, in which the two markers were amplified, were considered to have the *Ck-1* gene. Based on the markers, both hybrids have Dd genotype (Appendix A).

In the C3T, C4T, C6T, C7T, C10T and C11T populations, heterozygous resistant hybrids were identified for both markers. The C5T population (Catiguá MG2 x Arara) presented 10 hybrids that were homozygous-resistant for both markers.

The identification of hybrids containing the CBD-Sat235 marker (seven hybrids) alone and others containing the CBD-Sat207marker (20 hybrids) alone suggests that recombination occurred. These hybrids were not selected.

The best hybrids for the breeding program were selected based on genotypic and phenotypic data. Table 6 displays the best hybrids according to a selection pressure of 30% based on the selection index, mean rank position, and the resistance alleles (isolated or combined), and also includes the hybrid groups according to diversity analysis. The C2-10, C4-9 and C4-10 hybrids, selected by mean rank, showed pyramiding of the four resistance alleles against both *H. vastatrix* and *C. kahawae.*

## 4. Discussion

### 4.1. Crossing Certification

A crossing certification and contaminant detection in breeding populations are essential for the efficiency of plant breeding programs. This is particularly true for perennial and long cycle species such as coffee. Crossing certifications determine whether the genetic material selected for future generations is indeed the desired target since contaminating hybrids can affect gain in subsequent generations.

As *C. arabica* is an autogamous cleistogamous species, the hybridization process is manual, involving the emasculation of the flower to be used as a female parent. However, despite due care, self-fertilization may occur before emasculation and crossing, generating undesirable populations for the program. This self-fertilization contamination was the case with C1-3, C1-6, C1-7 and C1-8 coffee trees of the study population in this report. Self-fertilization contamination was previously reported in coffee [36]. Valencia et al. [36] analyzed the BA-124-Kf marker associated with the *S_H_3* gene in a F_1_ population and detected some coffee trees with no marker. They considered these plants contaminated and attributed this result to the fact that the respective hybridization had not occurred.

Hybrid coffees with genetic markers different from their parents were also detected in our work. This contamination may have occurred due to a crossing with pollen from other coffee trees than those selected as parents, or due to mixed seeds. In breeding programs, seed mixing can occur during the harvest period, as a large number of genotypes are handled. Thus, the use of molecular markers is essential to confirm plant identity. Yashitola et al. [37] used SSR markers to confirm the genetic purity of rice hybrids, concluding that this practice is considerably simpler than standard growth tests, which involve developing plants to maturity and evaluating their morphological and floral characteristics.

This study confirmed that the use of molecular markers for crossing certification is an important tool for recurrent selection breeding programs. This approach reduces labor, time, and financial resources by allowing the identification of undesirable plants.

### 4.2. Genetic Parameters and Components of Variance

The analysis of nine desired morphological and agronomic traits in coffee trees revealed genetic variability with a low magnitude between populations. This low variability was previously detected, especially in Arabica coffee cultivars [38,39], which were used in the crosses for the present study. These results highlight the importance of using MAS in Arabica coffee breeding populations to identify and increase genetic gain. The association of phenotypic and genotypic data allowed the selection of superior hybrids to compose the subsequent recurrent selection cycles and allowed hybrid selection with greater genetic variance (GV) and traits of economic relevance.

The populations studied exhibited low heritability for most analyzed traits. For yield (Y), a low value was expected as this trait is polygenic. However, vegetative vigor (Vig) showed the highest heritability (0.13), indicating the possibility of successful selection and the positive correlation of this trait with Y. According to Severino et al. [40], the selection of coffee with high Vig can increase Y since vegetative vigor has a direct effect on coffee productivity.

For data repeatability, most of the evaluated traits showed values below 10% in the studied populations. Resende [41] classified the repeatability as high (r ≥ 0.60), medium (0.30 ≤ r < 0.60), and low (r < 0.30). The low repeatability values observed indicate that a large number of repetitions are required to achieve a satisfactory determination value. The use of this coefficient increases the plant evaluation efficiency and reduces the time and labor [42], thereby decreasing costs and optimizing the breeding program.

On the other hand, according to Guerra et al. [43], the coefficients were used to determine environmental effects (c^2^parc) to quantify variability within blocks. The low values observed indicate a low level of environmental variation between the plots and within blocks, which suggests that the experimental design was adequate and that environmental homogeneity remained within the blocks.

Three analyzed traits were significant by the ANADEV approach; vegetative vigor (Vig), ripening fruit size (RFS), and cercosporiosis incidence (Cer). This significant effect indicates the existence of genetic variability in the progeny, confirming the genetic variance (GV) results.

### 4.3. Marker-Assisted Selection for H. vastatrix and C. kahawae Resistance

The populations evaluated in this study were formed by parent plants with traits of agronomic interest and alleles conferring resistance to the main diseases currently affecting the coffee crop. Thus, UFV 311-63 was used as a source for the *S_H_3* resistance gene. This coffee corresponds to an F_3_ plant derived from the Indian selections S.26 and S.31 backcrossed with the Kent or Coorg cultivars carrying the *S_H_2*, *S_H_3* and *S_H_5* alleles. The *S_H_3* gene has been associated with long-lasting resistance and confers resistance to a variety of *H. vastatrix* types. This gene, derived from *C. liberica,* was introduced into *C. arabica* using tetraploid coffee trees from the Indian selection and natural *C. arabica* × *C. liberica* [25]. UFV 311-63 was used in C2T, C4T and C12T crosses, justifying the identification of molecular markers associated with *S_H_3* only in coffee originated from crosses and the allocation to group 2 in the diversity analysis.

Although homozygous-resistant hybrids are ideal for generation progression, it was not possible to obtain this type of hybrid for the *S_H_3* gene because the UFV 311-63 parent plant was crossed with non-carrier coffee trees. However, coffee trees identified as heterozygous can be used in the program, because molecular markers can be used to select progeny with the desired target gene in subsequent generations.

Valencia et al. [36] evaluated eight F_1_ populations that contained the parental plants S288/23 and BA-2 (*S_H_3* gene carriers) in their genealogy using the BA-124-Kf marker alone. In the present study, four molecular markers linked to and flanking the *S_H_3* gene were used to improve selection accuracy. Hybrids with the four markers linked to the target gene only were selected as resistant hybrids. This strategy minimized the error that can be introduced by selecting hybrids that may have contained the marker but lost the allele by recombination.

Additional resistance sources were also used in the 12 crosses since the resulting populations contained various genes conferring resistance to other *H. vastatrix* types*,* in addition to hybrids with the *Ck-1* resistance gene to the *C. kahawae* pathogen. These alternative sources of resistance corresponded to HdT derivatives, which are carriers of the gene combinations *S_H_6*, *S_H_7*, *S_H_8*, *S_H_9* and *S_H_*? originating from *C. canephora* [23].

Molecular markers flanking the two QTL associated with resistance to the three different pathotypes were also analyzed within the populations to identify hybrids containing other *H. vastatrix* resistance alleles. The markers revealed that progeny from the C6T population (Appendix A) contained the two QTL, therefore expressing the BBC genotype (Appendix A). As they already presented pyramided alleles, the selection of these hybrids for future generations is promising. In the other populations, hybrids with the BBcc, Bbcc and bbC_ genotypes were identified (Appendix A), which did not have pyramided alleles but were also resistant and could be selected for the next generation. According to Pestana et al. [35], the resistance of coffee trees to races I and II and pathotype 001 *H. vastatrix* is conferred by two independent dominant loci. Thus, the presence of a dominant allele in one of these two loci is sufficient for the plant material to be resistant and thus selected for the next generation.

The populations were also analyzed for the presence of resistance allele to *C. kahawae*, as HdT derivatives were used in several crosses. The identification of hybrids containing the *C. kahawae* resistance gene via molecular markers was especially important for *C. arabica* breeding programs as it allows the selection of resistant plants in the absence of the pathogen. This strategy promotes the preventive management of this disease since CBD has not been reported in Latin America. However, there are concerns regarding the introduction of this pathogen to this region due to the damage caused by the disease in Africa. CBD is considered the main production limiting factor in coffee-producing countries in Africa [11].

The analysis of the two molecular markers flanking the *Ck-1* gene showed the possibility of recombination events between several hybrids, which could have resulted in the loss of the desired resistance alleles. As some hybrids contained CBD-Sat207 or CBD-Sat235 marker alone, hybrids of both markers for the *Ck-1* resistance locus were considered resistant. According to Frisch and Melchinger [44], the efficiency of MAS results from the distance and orientation of the markers in relation to the gene. To obtain high efficiency, it is essential that the distance between the markers and the genes is as short as possible and that the markers used are flanking the gene.

Considering the molecular markers associated with the four loci of resistance to the two diseases (CLR and CBD), 11 hybrids belonging to the C2T and C4T populations exhibited four pyramided resistance alleles. These hybrids corresponded to the AaBbC_Dd genotype (Appendix A). In addition, there were other 10 resistant homozygous hybrids for the *Ck-1* gene in the C5T population with the genotypes aaBbC_DD or aaBbccDD, as well as 39 other homozygous-resistant hybrids for LG2 QTL (C1T, C6T, C8T, C9T, C10T and C11T populations). Among the latter, seven hybrids from the C6T population carried LG5 QTL, corresponding to the genotype aaBbC_Dd (Appendix A). These hybrids are thus important parent plants for breeding programs that seek to develop cultivars with multiple and long-lasting resistance since their pyramided alleles can be readily used by the next generations.

Both genotypic and phenotypic data were used to select the best hybrids (Table 6). The phenotypic data with a selection index of 30% ranked 29 hybrids as superior. The genotypes of these superior hybrids were obtained using the corresponding molecular markers, allowing the observation of pyramided alleles of the four loci associated with *H. vastatrix* and *C. kahawae* resistance. The C4-10 hybrid, the first in the mean rank, presented resistance alleles from all four loci. The C2-10 and C4-9 hybrids also showed pyramided alleles originating from all four alleles and were in the mean rank at positions 13 and 24, respectively. The other hybrids showed different combinations of the four allele loci, which hence could be considered for subsequent crosses to ensure the pyramiding of the loci.

This study proposes a method to develop a recurrent selection program using molecular markers for the coffee breeding program (*C. arabica*), as presented in Figure 1. Such a program allows the formation of a base population followed by the production of new cultivars containing pyramided resistance alleles to *H. vastatrix* and *C. kahawae*, in addition to improving other traits of economic importance.

## 5. Conclusions

This study demonstrated the efficiency of using molecular markers as a tool for recurrent selection programs. Using molecular markers, 11 hybrids containing pyramided resistance alleles at the *S_H_3* locus and two resistance QTL to *H. vastatrix* types I, II and pathotype 001 were identified in addition to the *Ck-1* gene, which confers resistance to *C. kahawae.* The integration of data on genetic diversity and genetic parameters also allowed the selection of optimal crosses. Thus, this study not only selected the best hybrid but also identified which hybrids were allocated to different groups, producing more promising crosses in which greater genetic gains are expected.

## Figures and Tables

**Figure 1 genes-14-00189-f001:**
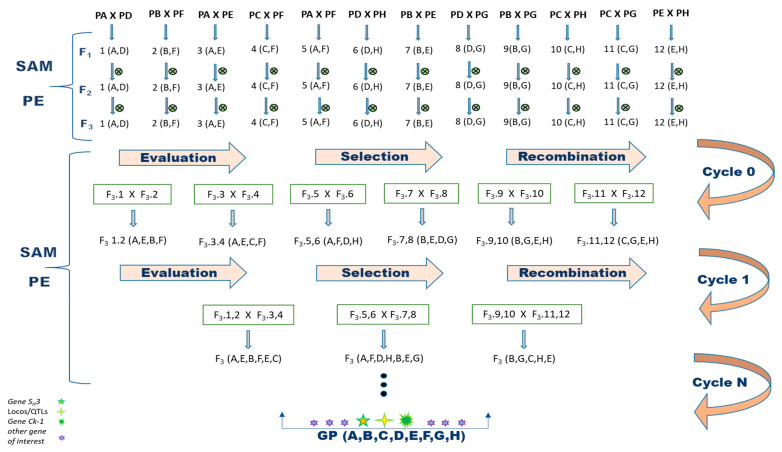
Proposal for the implementation of the recurrent selection method assisted by molecular marker aimed at pyramiding of *H. vastatrix* and *C. kahawae* resistance alleles in *C arabica.* Parents: PA = Paraíso MG H419-1; PB = Catiguá MG2; PC = Oeiras MG 6851; PD = H484-2-18-12; PE = UFV 311-63 plant; PF = Arara; PG = Acauã Novo and PH = Siriema; ⨂ = self-fertilization; SAM = selection assisted by molecular marker; PE = phenotypic evaluation; GP = gene pyramiding.

**Figure 2 genes-14-00189-f002:**
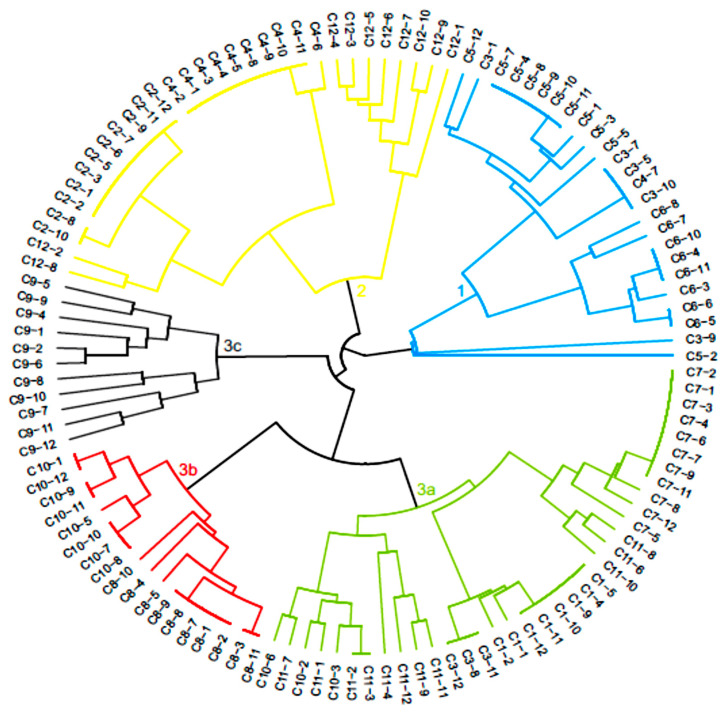
Dendrogram generated by the UPGMA method, based on the dissimilarity matrix of the arithmetic complement of the unweighted index of 121 *C. arabica* hybrids.

**Table 1 genes-14-00189-t001:** *C. arabica* hybrids obtained by crosses performed in a circulating diallel model between eight parents with important agronomic traits.

Crosses	Cross Code	Hybrid	Hybrid Code	Hybrid	Hybrid Code
	C1T	C1T-B1-P1-E1	C1-1	C1T-B3-P1-E1	C1-7
Paraíso MG H419-1		C1T-B1-P2-E1	C1-2	C1T-B3-P2-E1	C1-8
x		C1T-B1-P3-E1	C1-3	C1T-B3-P3-E1	C1-9
H484-2-18-2		C1T-B2-P1-E1	C1-4	C1T-B4-P1-E1	C1-10
		C1T-B2-P2-E1	C1-5	C1T-B4-P2-E1	C1-11
		C1T-B2-P3-E1	C1-6	C1T-B4-P3-E1	C1-12
	C2T	C2T-B1-P1-E1	C2-1	C2T-B3-P1-E1	C2-7
Paraíso MG H419-1		C2T-B1-P2-E1	C2-2	C2T-B3-P2-E1	C2-8
x		C2T-B1-P3-E1	C2-3	C2T-B3-P3-E1	C2-9
UFV 311-63 plant F_3_		C2T-B2-P1-E3	C2-4	C2T-B4-P1-E1	C2-10
		C2T-B2-P2-E1	C2-5	C2T-B4-P2-E1	C2-11
		C2T-B2-P3-E1	C2-6	C2T-B4-P3-E1	C2-12
	C3T	C3T-B1-P1-E1	C3-1	C3T-B3-P1-E1	C3-7
Paraíso MG H419-1		C3T-B1-P2-E3	C3-2	C3T-B3-P2-E1	C3-8
x		C3T-B1-P3-E3	C3-3	C3T-B3-P3-E3	C3-9
Arara		C3T-B2-P1-E3	C3-4	C3T-B4-P1-E1	C3-10
		C3T-B2-P2-E1	C3-5	C3T-B4-P2-E1	C3-11
		C3T-B2-P3-E1	C3-6	C3T-B4-P3-E1	C3-12
	C4T	C4T-B1-P1-E1	C4-1	C4T-B3-P1-E1	C4-7
Catiguá MG2		C4T-B1-P2-E1	C4-2	C4T-B3-P2-E1	C4-8
x		C4T-B1-P3-E1	C4-3	C4T-B3-P3-E1	C4-9
UFV 311-63 plant F_3_		C4T-B2-P1-E1	C4-4	C4T-B4-P1-E1	C4-10
		C4T-B2-P2-E1	C4-5	C4T-B4-P2-E1	C4-11
		C4T-B2-P3-E1	C4-6	C4T-B4-P29-E3	C4-12
	C5T	C5T-B1-P1-E1	C5-1	C5T-B3-P1-E1	C5-7
Catiguá MG2		C5T-B1-P2-E1	C5-2	C5T-B3-P2-E1	C5-8
x		C5T-B1-P3-E1	C5-3	C5T-B3-P3-E1	C5-9
Arara		C5T-B2-P1-E1	C5-4	C5T-B4-P1-E1	C5-10
		C5T-B2-P2-E1	C5-5	C5T-B4-P2-E1	C5-11
		C5T-B2-P3-E1	C5-6	C5T-B4-P3-E1	C5-12
	C6T	C6T-B1-P1-E1	C6-1	C6T-B3-P1-E1	C6-7
Catiguá MG2		C6T-B1-P2-E1	C6-2	C6T-B3-P2-E1	C6-8
x		C6T-B1-P3-E1	C6-3	C6T-B3-P3-E1	C6-9
Acauã Novo		C6T-B2-P1-E1	C6-4	C6T-B4-P1-E1	C6-10
		C6T-B2-P2-E1	C6-5	C6T-B4-P2-E1	C6-11
		C6T-B2-P3-E1	C6-6	C6T-B4-P3-E1	C6-12
	C7T	C7T-B1-P1-E1	C7-1	C7T-B3-P1-E1	C7-7
Oeiras MG 6851		C7T-B1-P2-E1	C7-2	C7T-B3-P2-E1	C7-8
x		C7T-B1-P3-E1	C7-3	C7T-B3-P3-E1	C7-9
Arara		C7T-B2-P1-E1	C7-4	C7T-B4-P1-E1	C7-10
		C7T-B2-P2-E1	C7-5	C7T-B4-P2-E1	C7-11
		C7T-B2-P3-E1	C7-6	C7T-B4-P3-E1	C7-12
	C8T	C8T-B1-P1-E1	C8-1	C8T-B3-P1-E1	C8-7
Oeiras MG 6851		C8T-B1-P2-E1	C8-2	C8T-B3-P2-E1	C8-8
x		C8T-B1-P3-E1	C8-3	C8T-B3-P3-E1	C8-9
Acauã Novo		C8T-B2-P1-E1	C8-4	C8T-B4-P1-E1	C8-10
		C8T-B2-P2-E1	C8-5	C8T-B4-P2-E1	C8-11
		C8T-B2-P3-E1	C8-6	C8T-B4-P3-E1	C8-12
	C9T	C9T-B1-P1-E1	C9-1	C9T-B3-P1-E1	C9-7
Oeiras MG 6851		C9T-B1-P2-E1	C9-2	C9T-B3-P2-E1	C9-8
x		C9T-B1-P3-E1	C9-3	C9T-B3-P3-E1	C9-9
Siriema		C9T-B2-P1-E1	C9-4	C9T-B4-P1-E1	C9-10
		C9T-B2-P2-E1	C9-5	C9T-B4-P2-E1	C9-11
		C9T-B2-P3-E1	C9-6	C9T-B4-P3-E1	C9-12
	C10T	C10T-B1-P1-E1	C10-1	C10T-B3-P1-E1	C10-7
H484-2-18-2		C10T-B1-P2-E1	C10-2	C10T-B3-P2-E1	C10-8
x		C10T-B1-P3-E1	C10-3	C10T-B3-P3-E1	C10-9
Acauã Novo		C10T-B2-P1-E1	C10-4	C10T-B4-P1-E1	C10-10
		C10T-B2-P2-E1	C10-5	C10T-B4-P2-E1	C10-11
		C10T-B2-P3-E1	C10-6	C10T-B4-P3-E1	C10-12
	C11T	C11T-B1-P1-E1	C11-1	C11T-B3-P1-E1	C11-7
H484-2-18-12		C11T-B1-P2-E1	C11-2	C11T-B3-P2-E1	C11-8
x		C11T-B1-P3-E1	C11-3	C11T-B3-P3-E1	C11-9
Siriema		C11T-B2-P1-E1	C11-4	C11T-B4-P1-E1	C11-10
		C11T-B2-P2-E1	C11-5	C11T-B4-P2-E1	C11-11
		C11T-B2-P3-E1	C11-6	C11T-B4-P3-E1	C11-12
	C12T	C12T-B1-P1-E1	C12-1	C12T-B3-P1-E1	C12-7
UFV 311-63 plant F_3_		C12T-B1-P2-E1	C12-2	C12T-B3-P2-E1	C12-8
x		C12T-B1-P3-E1	C12-3	C12T-B3-P3-E1	C12-9
Siriema		C12T-B2-P1-E1	C12-4	C12T-B4-P1-E1	C12-10
		C12T-B2-P2-E1	C12-5	C12T-B4-P2-E1	C12-11
		C12T-B2-P3-E1	C12-6	C12T-B4-P3-E1	C12-12

**Table 2 genes-14-00189-t002:** Description of the phenotypic traits evaluated in the years 2015 and 2016 in field.

Traits		Evaluation Description
Vegetative vigor	(Vig)	Score scale ranging from 1 to 10: 1 = fully depauperate (depleted) plants; 10 = plant with maximum vegetative vigor.
Yield	(Y)	Liters of freshly harvested cherries per plant.
Fruit ripening uniformity	(FRU)	Score scale ranging from 1 to 4: 1 = uniform; 2 = moderately uniform; 3 = moderately not uniform; 4 = not uniform.
Fruit size at ripening	(RFS)	Fruit size at ripening (Score scale from 1 to 3). 1 = small, 2 = medium, 3 = large fruits.
Rust incidence	(Rus)	Score scale ranging from 1 to 5: 1 = absence of pustules and hypersensitivity reactions; 2 = few leaves with spore-free pustules (“flecks”) and with hypersensitivity reactions; 3 = few pustules per leaf with production of spores; 4 = average amount of pustules per leaf, distributed in the plant with high spore production; 5 = high amount of pustules with high spore production and defoliation of the plant (plants with score 1 to 2 = resistant; 3 to 5 = susceptible).
Cercosporiosis incidence	(Cer)	Score scale ranging from 1 to 5: 1 = without symptoms; 5 = highly susceptible.
Leaf miner infestation	(LM)	Score scale ranging from 1 to 5: 1 = without leaf miner; 5 = highly infested.
No of pairs of plagiotropic branches	(NPB)	Number of pairs of plagiotropic branches on the main stem.
Plant height (cm)	(PH)	Measured in the orthotropic branch (from the soil surface to the final branch growth point).

**Table 3 genes-14-00189-t003:** Estimate of genetic parameters obtained by mixed model analyses (REML/BLUP) for nine morpho-agronomic traits in F1 progenies of *C. arabica*.

Genetic Parameters	Vig	Y	FRU	RFS	Rus	Cer	LM	NPB	PH
Vg	0.19	0.14	0.01	0.02	0.00	0.04	0.00	0.01	8.22
Vparc	0.08	0.02	0.02	0.00	0.00	0.00	0.00	0.14	1.94
Vperm	0.08	0.02	0.02	0.00	0.00	0.00	0.00	0.14	1.94
Ve	1.09	3.77	0.30	0.19	0.25	0.54	0.20	29.30	432.50
Vf	1.45	3.94	0.34	0.22	0.26	0.59	0.20	29.60	444.61
h^2^g	0.13	0.04	0.02	0.09	0.02	0.07	0.00	0.00	0.02
r	0.25	0.04	0.12	0.12	0.03	0.08	0.02	0.01	0.03
c^2^parc	0.06	0.00	0.05	0.01	0.00	0.00	0.01	0.00	0.00
c^2^perm	0.06	0.00	0.05	0.01	0.00	0.00	0.01	0.00	0.00
Mean	7.64	3.02	2.97	2.15	1.58	2.32	1.17	2.41	0.00

Vg = genotypic variance; Vparc = environmental variance between plots; Vperm = permanent environment variance; Ve = residual variance; Vf = phenotypic variance individual; h^2^g = heritability of individual plants in the broad sense; r = repeatability; c^2^parc = coefficient of determination plots; c^2^perm = permanent coefficient of determination; Mean = overall mean of the experiment; Vig = vegetative vigor; Y= yield; FRU = fruit ripening uniformity; RFS = ripening fruits size; Rus = rust incidence; Cer = cercosporiosis incidence; LM = leaf miner infestation; NPB = n° of plagiotropic branches pair; PH, plant height.

**Table 4 genes-14-00189-t004:** Analysis of deviance (ANADEV) and significance of each effect by likelihood ratio test (LRT) considering the joint analysis of morphological traits measured in 2015 and 2016, in *C. arabica* F_1_ progenies.

Variables	DEVIANCE		LRT (X^2^)
	Complete Model	Effect between Hybrids	Effect between Hybrids
Vig	329.38	340.99	11.61 **
Y	551.65	553.19	1.54 ^ns^
FRU	11.63	12.29	0.66 ^ns^
RFS	−87.97	−78.9	9.07 **
Rus	−48.14	−47.46	0.68 ^ns^
Cer	136.86	142.79	5.93 *
LM	−99.12	−99.11	0.01 ^ns^
NPB	1026.73	1026.73	0 ^ns^
PH	1638.24	1638.91	0.67 ^ns^

ns = not significant, ** = significant at the 1% probability, * = significant at 5% probability, using the chi-square test with 1 degree of freedom. Vig = vegetative vigor; Y= yield; FRU = fruit ripening uniformity; RFS = ripening fruits size; Rus = rust incidence; Cer = cercosporiosis incidence; LM = leaf miner infestation; NPB = n° of plagiotropic branches pair; PH, plant height.

**Table 5 genes-14-00189-t005:** Analysis of molecular variance (AMOVA) for 12 hybrid F1 populations using SSR markers.

Variation Source	Degree of Freedom	Sum of Squares	Medium Squares	Variance Component	% of Total Component Variance
Among populations	11	118108	10737	0.1033	75.513
Within populations	109	36515	0.0335	0.0335	24.487
Total	120	154623	0.1289		

Statistic ØST = 0.7551.

**Table 6 genes-14-00189-t006:** The best hybrids selected by the mean rank, with the selection gains (%), genetic diversity and presence of resistant alleles to *H. vastatrix* and *C. kahawae*, as inferred by molecular markers.

Hybrid Code	Selection Gain %	Dendrogram Group	Resistance Genes			
			S_H_3	* LG2	* LG5	Ck-1
C4-10	92.8	2	Aa	Bb	C_	Dd
C3-9	84.0	1	aa	Bb	C_	Dd
C2-12	84.0	2	Aa	Bb	cc	dd
C3-8	80.7	3a	aa	Bb	C_	Dd
C9-4	75.8	3c	aa	BB	cc	dd
C3-12	69.7	3a	aa	Bb	cc	Dd
C8-5	62.5	3b	aa	BB	cc	dd
C12-1	56.5	2	Aa	Bb	cc	dd
C10-10	56.5	3b	aa	BB	cc	Dd
C12-6	52.6	2	Aa	bb	cc	dd
C9-6	51.5	3c	aa	Bb	cc	dd
C4-6	51.0	2	Aa	Bb	cc	dd
**C2-10**	**49.9**	**2**	**Aa**	**Bb**	**C_**	**Dd**
C8-7	49.3	3b	aa	BB	cc	dd
C10-9	49.3	3b	aa	BB	cc	Dd
C11-7	48.2	3a	aa	Bb	cc	Dd
C5-8	47.1	1	aa	Bb	C_	DD
C7-3	46.0	3a	aa	bb	C_	Dd
C6-11	46.0	1	aa	BB	C_	Dd
C12-9	43.8	2	Aa	Bb	cc	dd
C7-2	42.7	3a	aa	bb	cc	Dd
C8-2	42.7	3b	aa	BB	cc	dd
C3-10	42.1	1	aa	Bb	C_	Dd
**C4-9**	**41.6**	**2**	**Aa**	**Bb**	**C_**	**Dd**
C9-5	41.0	3c	aa	BB	cc	dd
C2-5	39.4	2	Aa	Bb	cc	dd
C4-7	39.4	1	aa	Bb	C_	Dd
C5-5	38.3	1	aa	Bb	C_	DD
C1-2	37.7	3a	aa	BB	cc	Dd

Bold = hybrids with pyramidation of resistance alleles to *H. vastatrix* and *C. kahawae*. * LG2 and LG5 = loci/QTL that correspond to major genes that confer resistance to races I, II and pathotype 001 of *H. vastatrix*.

## Data Availability

The data presented in this study are available in Appendix A.

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
