# Peer review of "Marker-Assisted Recurrent Selection for Pyramiding Leaf Rust and Coffee Berry Disease Resistance Alleles in Coffea arabica L."

_genes, 2023, doi:10.3390/genes14010189_

Round 1

Reviewer 1 Report

The manuscript sound good and attractive. I have some minor concern as bellow.

1. The authors have pyramid the genes for two diseases (CLR and CBD), but the mentioned the main diseases in the title. The title may be revised. Please put the name of these two diseases in the title. It may be “Marker assisted recurrent selection for pyramiding CLR and CBD diseases in coffee.”

2. Please write the scientific name of the crop in the title.

3. Line-17. The scientific name should be italic.

4. Line-21. Expand the CLR and CBD.

5. Line-48. Start the sentence with “Pyramiding …………..”.

6. Line-58. Put the full stop after completion of the sentence.

7. Line-61. Better to write the name of other resistant genes.

8. Line-70. “Incorporation” may be replaced with “introduction”.

9. Line-97. The scientific name should be italic.

10. In table-1. “Code” will be replaced with “hybrid code”.

11. In table-2. “Ripening fruit size” may be write as “Fruit size at ripening”.

12. Please correct “No of pair”.

13. Line-123. Citation will not be in bold.

14. Line-144. Put the full stop.

15. Line-149. “initiated with” replaced with “set as initial denaturation………………”.

16. Line-164. “Presents” may be replaced with “comprised”.

17. In section 2.4, 2.5, 2.6, 2.7. The PCR conditioning was presented separately. It is almost repetition. Try to compile in one section may be in the last one of in first one.

18.  In figure-1. what is PA, PC, PD. Please write as footnote.

19. In table-4. “0,66” may be “0.66”.

20. Check the references. Number are duplicated. Correct it. Many errors had seen in the reference section. Like scientific names of the crops should be italic. Please check the journal names. 

21. I also highlighted the manuscript. 

Author Response

Dear

We are grateful for the rapid revision of our manuscript entitled “Marker-assisted recurrent selection applied in pyramiding resistance alleles for the main coffee diseases” to be considered for publication in Molecular Genetics and Genomics Sections, Genes Journal. We have carefully considered the reviewers’ constructive comments and incorporated all the suggestions. Point-by-point response is below:

  1. The authors have pyramid the genes for two diseases (CLR and CBD), but the mentioned the main diseases in the title. The title may be revised. Please put the name of these two diseases in the title. It may be “Marker assisted recurrent selection for pyramiding CLR and CBD diseases in coffee.”

Response: The title modified as per the suggestion

  1. Please write the scientific name of the crop in the title.

Response: Scientific name included in the title

  1. Line-17. The scientific name should be italic.

Response: It has been italicized

  1. Line-21. Expand the CLR and CBD.

Response: The abbreviations have been expanded

  1. Line-48. Start the sentence with “Pyramiding …………..”.

Response: The sentence has been modified accordingly.

  1. Line-58. Put the full stop after completion of the sentence.

Response: Full stop has been put

  1. Line-61. Better to write the name of other resistant genes.

Response: Have been written the names of the other resistance genes “CC-NBS-LRR gene and HdT_LRR_RLK2”

  1. Line-70. “Incorporation” may be replaced with “introduction”.

Response: ‘Incorporation’ has been replaced by ‘introduction’

  1. Line-97. The scientific name should be italic.

Response: Italicized

  1. In table-1. “Code” will be replaced with “hybrid code”.

Response: corrected

  1. In table-2. “Ripening fruit size” may be write as “Fruit size at ripening”.

Response: Corrected

  1. Please correct “No of pair”.

Response: corrected

  1. Line-123. Citation will not be in bold.

Response: corrected

  1. Line-144. Put the full stop.

Response:  full stop included

  1. Line-149. “initiated with” replaced with “set as initial denaturation………………”.

Response: the sentence was edited

  1. 16. Line-164. “Presents” may be replaced with “comprised”.

Response: replacement was made

  1. In section 2.4, 2.5, 2.6, 2.7. The PCR conditioning was presented separately. It is almost repetition. Try to compile in one section may be in the last one of in first one.

Response: Thank you for the comment, however there are some differences in the concentration of some reagents and/or volume of the final PCR reaction in four of the sections. These differences are important for repeating the reaction correctly.

  1. In figure-1. what is PA, PC, PD. Please write as footnote.

Response: corrected

  1. In table-4. “0,66” may be “0.66”.

Response: corrected

  1. Check the references. Number are duplicated. Correct it. Many errors had seen in the reference section. Like scientific names of the crops should be italic. Please check the journal names. 

Response: corrected

  1. I also highlighted the manuscript.

Response: Thank you.

Reviewer 2 Report

In this study, 144 Arabica coffee genotypes with morphological and agronomic features from crossing of 8 parent plants were investigated. This approach led to more promising crossings with anticipated higher genetic gains by not only selecting the best hybrid but also identifying which hybrids belonged to which groups. However, the following list of minor revisions is included for the reader's convenience.

1.    Certain phenotypic features, including Vig, FRU, RFS, Rus, Cer, and LM, could not be quantitatively assessed. Please provide some figures as examples to show the various scores.

2.    Please provide the complete species name of PA-PH for Fig. 1.

3.    Each reference in the References section has two numbers in front of them.

4.    "ηg" should be replaced by "ng" in lines 145 and 202.

Author Response

Dear

We are grateful for the rapid revision of our manuscript entitled “Marker-assisted recurrent selection applied in pyramiding resistance alleles for the main coffee diseases” to be considered for publication in Molecular Genetics and Genomics Sections, Genes Journal. We have carefully considered the reviewers’ constructive comments and incorporated all the suggestions. Point-by-point response is below:

  1. Certain phenotypic features, including Vig, FRU, RFS, Rus, Cer, and LM, could not be quantitatively assessed. Please provide some figures as examples to show the various scores.

Response: Thank you for the comments. The variation score of these traits is in Table 3, we made correction on the morpho-agronomic traits names to make it clear.

  1. Please provide the complete species name of PA-PH for Fig. 1.

Response: Corrected

  1. Each reference in the References section has two numbers in front of them.

Response: Corrected

  1. "ηg" should be replaced by "ng" in lines 145 and 202.

Response: Corrected
